# DATT: Deep Adaptive Trajectory Tracking for Quadrotor Control

**Kevin Huang**
University of Washington
kehuang@cs.washington.edu

**Rwik Rana**
University of Washington
rwik2000@uw.edu

**Alexander Spitzer**
University of Washington
spitzer@cs.washington.edu

**Guanya Shi**
Carnegie Mellon University
guanyas@andrew.cmu.edu

**Byron Boots**
University of Washington
bboots@cs.washington.edu

**Abstract:** Precise arbitrary trajectory tracking for quadrotors is challenging due to unknown nonlinear dynamics, trajectory infeasibility, and actuation limits. To tackle these challenges, we present Deep Adaptive Trajectory Tracking (DATT), a learning-based approach that can precisely track arbitrary, potentially infeasible trajectories in the presence of large disturbances in the real world. DATT builds on a novel feedforward-feedback-adaptive control structure trained in simulation using reinforcement learning. When deployed on real hardware, DATT is augmented with a disturbance estimator using $\mathcal{L}_1$ adaptive control in closed-loop, without any fine-tuning. DATT significantly outperforms competitive adaptive nonlinear and model predictive controllers for both feasible smooth and infeasible trajectories in unsteady wind fields, including challenging scenarios where baselines completely fail. Moreover, DATT can efficiently run online with an inference time less than $3.2\,\mathrm{ms}$, less than 1/4 of the adaptive nonlinear model predictive control baseline[1].

**Keywords:** Quadrotor, Reinforcement Learning, Adaptive Control

## 1 Introduction

Executing precise and agile flight maneuvers is important for the ongoing commoditization of unmanned aerial vehicles (UAVs), in applications such as drone delivery, rescue and search, and urban air mobility. In particular, accurately following *arbitrary trajectories* with quadrotors is among the most notable challenges to precise flight control for the following reasons. First, quadrotor dynamics are highly nonlinear and underactuated, and often hard to model due to unknown system parameters (e.g., motor characteristics) and uncertain environments (e.g., complex aerodynamics from unknown wind gusts). Second, aggressive trajectories demand operating at the limits of system performance, requiring awareness and proper handling of actuation constraints, especially for quadrotors with small thrust-to-weight ratios. Finally, the arbitrary desired trajectory might not be *dynamically feasible* (i.e., impossible to stay on such a trajectory), which necessities long-horizon reasoning and optimization in real-time. For instance, to stay close to the five-star trajectory in Fig. 1, which is infeasible due to the sharp changes of direction, the quadrotor must predict, plan, and react online before the sharp turns.

Traditionally, there are two commonly deployed control strategies for accurate trajectory following with quadrotors: nonlinear control based on differential flatness and model predictive control

---

[1]Videos and demonstrations in https://sites.google.com/view/deep-adaptive-traj-tracking and code in https://github.com/KevinHuang8/DATT.

7th Conference on Robot Learning (CoRL 2023), Atlanta, USA.

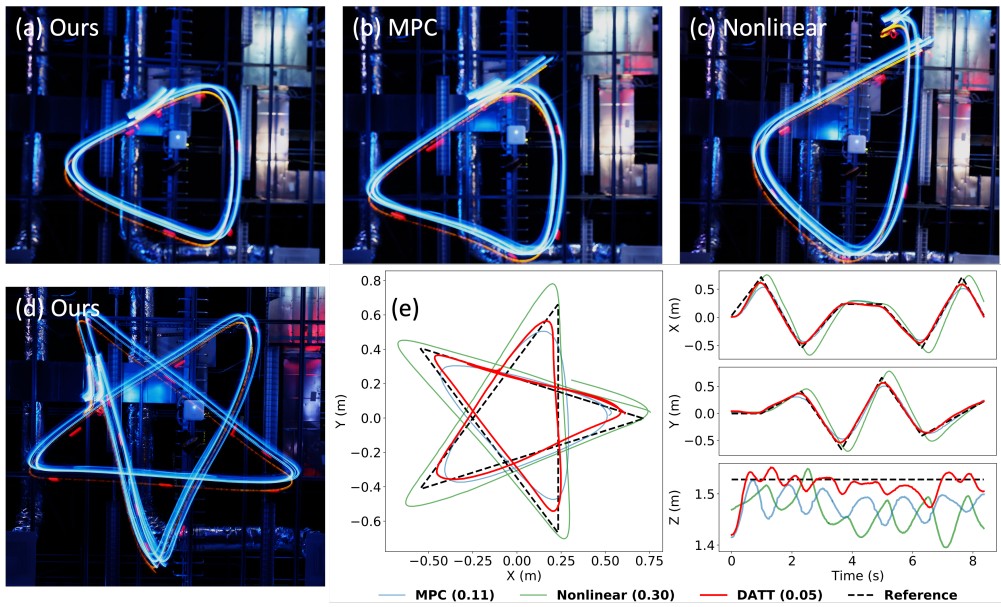

Figure 1: Trajectory visualizations for example infeasible trajectories. (a-c) Long-exposure photos of different methods for an equilateral triangle reference trajectory. (d) Long-exposure photo of our method for a five-pointed star reference trajectory. (e) Quantitative comparisons between our approach and baselines for the five-pointed star. Numbers indicate the tracking error in meters.

(MPC). However, nonlinear control methods, despite their proven stability and efficiency, are constrained to differentially flat trajectories (i.e., smooth trajectories with bounded velocity, acceleration, jerk, and snap) satisfying actuation constraints [1, 2, 3]. On the other hand, MPC approaches can potentially incorporate constraints and non-smooth arbitrary trajectories [4, 5], but their performances heavily rely on the accuracy of the model and the optimality of the solver for the underlying nonconvex optimization problems, which could also be expensive to run online.

Reinforcement learning (RL) has shown its potential flexibility and efficiency in trajectory tracking problems [6, 7, 8]. However, most existing works focus on tracking smooth trajectories in stationary environments. In this work, we aim to design an RL-based flight controller that can (1) follow feasible trajectories as accurately as traditional nonlinear controllers and MPC approaches; (2) accurately follow arbitrary infeasible and dynamic trajectories to the limits of the hardware platform; and (3) adapt to unknown system parameters and uncertain environments online. Our contributions are:

- We propose DATT, a novel feedforward-feedback-adaptive policy architecture and training pipeline for RL-based controllers to track arbitrary trajectories. In training, this policy is conditioned on ground-truth translational disturbance in a simulator, and such a disturbance is estimated in real using $\mathcal{L}_1$ adaptive control in closed-loop;

- On a real, commercially available, lightweight, and open-sourced quadrotor platform (Crazyflie 2.1 with upgraded motors), we show that our approach can track feasible smooth trajectories with 27%-38% smaller errors than adaptive nonlinear or adaptive MPC baselines. Moreover, our approach can effectively track infeasible trajectories where the nonlinear baseline completely fails, with a 39% smaller error than MPC and 1/4th the computational time;

- On the real quadrotor platform, we show that our approach can adapt zero-shot to unseen turbulent wind fields with an extra cardboard drag plate for both smooth desired trajectories and infeasible trajectories. Specifically, for smooth trajectories, our method achieves up to 22% smaller errors than the state-of-the-art adaptive nonlinear control method. In the most challenging scenario (infeasible trajectories with wind and drag plate), our method significantly outperforms the adaptive MPC approach with 15% less error and 1/4th of the computation time.

## 2 Problem Statement and Related Work

### 2.1 Problem Statement

In this paper, we let $\dot{x}$ denote the derivative of a continuous variable $x$ regarding time. We consider the following quadrotor dynamics:

$$\dot{p} = v, \qquad\qquad m\dot{v} = mg + Re_3 f_\Sigma + d \qquad\qquad (1a)$$

$$\dot{R} = RS(\omega), \qquad\qquad J\dot{\omega} = J\omega \times \omega + \tau, \qquad\qquad (1b)$$

where $p, v, g \in \mathbb{R}^3$ are position, velocity, and gravity vectors in the world frame, $R \in \mathrm{SO}(3)$ is the attitude rotation matrix, $\omega \in R^3$ is the angular velocity in the body frame, $m, J$ are mass and inertia matrix, $e_3 = [0; 0; 1]$, and $S(\cdot) : \mathbb{R}^3 \to \mathrm{so}(3)$ maps a vector to its skew-symmetric matrix form. Moreover, $d$ is the time-variant translational disturbance, which includes parameter mismatch (e.g., mass error) and environmental perturbation (e.g., wind perturbation) [9, 10, 11, 12]. The control input is the total thrust $f_\Sigma$ and the torque $\tau$ in the body frame. For quadrotors, there is a linear invertible actuation matrix between $[f_\Sigma; \tau]$ and four motor speeds.

We let $x_t$ denote the temporal discretization of $x$ at time step $t \in \mathbb{Z}_+$. In this work, we focus on the 3-D trajectory tracking problem with the desired trajectory $p_1^d, p_2^d, \cdots, p_T^d$, with average tracking error as the performance metric: $\frac{1}{T}\sum_{t=1}^T \|p_t - p_t^d\|$. We do not have any assumptions on the desired trajectory $p^d$. In particular, $p^d$ is not necessarily differentiable or smooth.

### 2.2 Differential Flatness

The differential flatness property of quadrotors allows efficient generation of control inputs to follow smooth trajectories [1, 5]. Differential flatness has been extended to account for unknown linear disturbances [3], learned nonlinear disturbances [13], and also to deal with the singularities associated with pitching and rolling past 90 degrees [14]. While differential-flatness-based methods can show impressive performance for smooth and aggressive trajectories, they struggle with nondifferentiable trajectories or trajectories that require reasoning about actuation constraints.

### 2.3 Model Predictive Control (MPC)

MPC is a widely used optimal control approach that online optimizes control inputs over a finite time horizon, considering system dynamics and constraints [15, 16].

Model Predictive Path Integral Control (MPPI) [4, 17] is a sampling-based MPC incorporating path integral control formulation and stochastic sampling. Unlike deterministic optimization, MPPI employs a stochastic optimization approach where control sequences are sampled from a distribution. These samples are then evaluated based on a cost function, and the distribution is iteratively updated to improve control performance. Recently MPPI has been applied to quadrotor control [18, 19].

Gradient-based nonlinear MPC techniques have been widely used for rotary-winged-based flying robots or drones. Hanover et al. [12] and Sun et al. [5] have shown good performance of nonlinear MPC in agile trajectory tracking of drones and adaptation to external perturbations. Moreover, these techniques are being used for vision-based agile maneuvers of drones [20, 7].

However, for either sampling-based or gradient-based MPC, the control performance heavily relies on the optimality of the optimizer for the underlying nonconvex problems. Generally speaking, MPC-based approaches require much more computing than differential-flatness-based methods [5]. Moreover, MPC's robustness and adaptability for infeasible trajectories remain unclear since existing works consider smooth trajectory tracking. In this paper, we implemented MPPI [4] and $\mathcal{L}_1$ augmented MPPI [18] for our baselines.

### 2.4 Adaptive Control and Disturbance Estimation

Adaptive controllers aim to improve control performance through online estimation of unknown system parameters in closed-loop. For quadrotors, adaptive controllers typically estimate a three-dimensional force disturbance $d$ [21, 10, 22, 23, 18]. Most recently, $\mathcal{L}_1$ adaptive control for quadrotors [11] has been shown to improve trajectory tracking performance in the presence of complex and time-varying disturbances such as sloshing payloads and mismatched propellers. Recently, deep-learning-based adaptive flight controllers have also emerged [10, 24, 25].

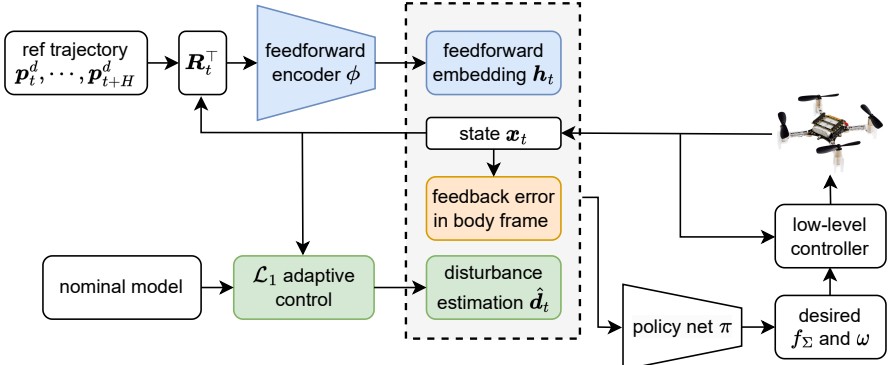

Figure 2: Algorithm Overview. Blue, yellow, and green blocks represent feedforward, feedback, and adaptation modules respectively. In training the policy has access to the true disturbance $d$ whereas in real we use $\mathcal{L}_1$ adaptive control to get the disturbance estimation $\hat{d}$ in closed-loop.

Learning dynamical models is a common technique to improve quadrotor trajectory tracking performance [9, 26, 27, 28] and can provide more accurate disturbance estimates than purely reactive adaptive control, due to the model of the disturbance over the state and control space. In this work, we use the disturbance estimation from $\mathcal{L}_1$ adaptive control, but we note that our method can leverage any disturbance estimation or model learning techniques.

In particular, Rapid Motor Adaptation (RMA) is a supervised learning-based approach that aims to predict environmental parameters using a history of state-action pairs, which are then inputted to the controller [29]. This approach has been shown to work for real legged-robots, but we find that it can be susceptible to domain shift during sim2real transfer on drones.

### 2.5 Reinforcement Learning for Quadrotor Control

Reinforcement learning for quadrotor stabilization is studied in [6, 30, 24]. Molchanov et al. [30] uses domain randomization to show policy transfer between multiple quadrotors. Kaufmann et al. [31] compares three different policy formulations for quadrotor trajectory tracking and finds that outputting body thrust and body rates outperforms outputting desired linear velocities and individual rotor thrusts. [31] only focuses on feasible trajectories while in this work, we aim to track infeasible trajectories as accurately as possible. Simulation-based learning with imitation learning to an expert MPC controller is used to generate acrobatic maneuvers in [7]. In this work, we focus on trajectories and environments for which obtaining an accurate expert even in simulation is difficult or expensive and thus use reinforcement learning to learn the controller.

## 3 Methods

### 3.1 Algorithm Overview

A high-level overview of DATT is given in Fig. 2. Using model-free RL, DATT learns a neural network quadrotor controller $\pi$ capable of tracking arbitrary reference trajectories, including infeasible trajectories, while being able to adapt to various environmental disturbances, even those unseen during training. We condition our policy on a learned *feedforward embedding* $h$, which encodes the desired reference trajectory, in the body frame, over a fixed time horizon, as well as the force disturbance $d$ in Eq. (1).

The state $x_t$ consists of the position $p$, the velocity $v$, and the orientation $R$, represented as a quaternion $q$. We convert $p, v$ to the body frame and input them to $\pi$. Our policy controller outputs $u$ which includes the desired total thrust $f_{\Sigma,\mathrm{des}}$, and the desired body rates $\omega_{\mathrm{des}}$. In summary, our controller functions as follows:

$$h_t = \phi(R_t^\top(p_t - p_t^d)), \ldots, R_t^\top(p_t - p_{t+H}^d)) \tag{2a}$$

$$u_t = \pi(R_t^\top p_t, R_t^\top v_t, q_t, h_t, R_t^\top(p_t - p_t^d), d_t) \tag{2b}$$

We define the expected reward for our policy conditioned on the reference trajectory as follows:

$$J(\boldsymbol{\pi}|\boldsymbol{p}_{t:t+H}^d) = \mathbb{E}_{(\boldsymbol{x},\boldsymbol{u})\sim\boldsymbol{\pi}}\left[\sum_{t=0}^{\infty} r(\boldsymbol{x}_t,\boldsymbol{u}_t|\boldsymbol{p}_{t:t+H}^d)\right] \tag{3a}$$

$$r(\boldsymbol{x}_t,\boldsymbol{u}_t|\boldsymbol{p}_{t:t+H}^d) = \|\boldsymbol{p}_t - \boldsymbol{p}_t^d\| + 0.5\|\psi_t\| + 0.1\|\boldsymbol{v}_t\| \tag{3b}$$

$\psi_t$ denotes the yaw of the drone. The reward function optimizes for accurate position and yaw tracking, with a small velocity regularization penalty. $\boldsymbol{\pi}$ and $\phi$ are jointly optimized with respect to $J$ using the Proximal Policy Optimization (PPO) algorithm [32].

## 3.2 Arbitrary Trajectory Tracking

Classical controllers, such as differential-flatness controllers, rely on higher-order position derivatives of the reference trajectory for accurate tracking (velocity, acceleration, jerk, and snap), which are needed for incorporating future information about the reference, i.e., feedforward control. However, arbitrary trajectories can have undefined higher order derivatives, and exact tracking may not be feasible. With RL, a controller can be learned to optimally track an arbitrary reference trajectory, given just the desired future positions $\boldsymbol{p}_t^d$. Thus, we input just the desired positions, in the body-frame, into a feedforward encoder $\phi$, which learns the feedforward embedding that contains the information of the desired future reference positions. For simplicity, we assume the desired yaw for all trajectories is zero. The reference positions are provided evenly spaced from the current time $t$ to the feedfoward horizon $t + H$, and are transformed into the body frame.

## 3.3 Adaptation to Disturbance

During training in simulation, we add a random time-varying force perturbation $\boldsymbol{d}$ to the environment. We use $\mathcal{L}_1$ adaptive control [11, 33] to estimate $\boldsymbol{d}$, which is directly passed into our policy network during both training and inference. $\mathcal{L}_1$ adaptive control first builds a closed-loop estimator to compute the difference between the predicted and true disturbance, and then uses a low pass filter to update the prediction. The adaptation law is given by:

$$\dot{\hat{\boldsymbol{v}}} = \boldsymbol{g} + \boldsymbol{R}\boldsymbol{e}_3 f_\Sigma/m + \hat{\boldsymbol{d}}/m + \boldsymbol{A}_s(\hat{\boldsymbol{v}} - \boldsymbol{v}) \tag{4a}$$

$$\hat{\boldsymbol{d}}_{\text{new}} = -(e^{\boldsymbol{A}_s dt} - \boldsymbol{I})^{-1}\boldsymbol{A}_s e^{\boldsymbol{A}_s dt}(\hat{\boldsymbol{v}} - \boldsymbol{v}) \tag{4b}$$

$$\hat{\boldsymbol{d}} \leftarrow \text{low pass filter}(\hat{\boldsymbol{d}}, \hat{\boldsymbol{d}}_{\text{new}}) \tag{4c}$$

where $\boldsymbol{A}_s$ is a Hurwitz matrix, $dt$ is the discretization step length and $\hat{\boldsymbol{v}}$ is the velocity prediction. Generally speaking, (4a) is a velocity predictor using the estimated disturbance $\hat{\boldsymbol{d}}$, and (4b) and (4c) update and filter $\hat{\boldsymbol{d}}$. Compared to other sim-to-real techniques such as domain randomization [30] and student-teacher adaptation [24], the adaptive-control-based disturbance adaptation method in DATT tends to be more reactive and robust, thanks to the closed-loop nature and provable stability and convergence of $\mathcal{L}_1$ adaptive control.

We note that DATT provides a general framework for adaptive control. Other methods to estimate $\hat{\boldsymbol{d}}$, for example RMA, can easily be used instead, but we found them to be less robust than $\mathcal{L}_1$ adaptive control. We compare against an RMA baseline in our experiments.

# 4 Experiments

## 4.1 Simulation and Training

Training is done in a custom quadrotor simulator that implements (1) using on-manifold integration, with body thrust and angular velocity as the inputs to the system. In order to convert the desired body thrust $f_{\Sigma,\text{des}}$ and body rate $\boldsymbol{\omega}_{\text{des}}$ output from the controller to the actual thrust and body rate for the drone in simulation, we use a first-order time delay model:

$$\boldsymbol{\omega}_t = \boldsymbol{\omega}_{t-1} + k(\boldsymbol{\omega}_{\text{des}} - \boldsymbol{\omega}_{t-1}) \tag{5a}$$

$$f_{\Sigma,t} = f_{\Sigma,t-1} + k(f_{\Sigma,\text{des}} - f_{\Sigma,t-1}) \tag{5b}$$

We set $k$ to a fixed value of $0.4$, which we found worked well on the real drone. In practice, the algorithm generalizes well to a large range of $k$, even when training on fixed $k$. Our simulator effectively runs at $50\,\text{Hz}$, with $dt = 0.02$ for each simulation step.

We train across a series of xy-planar smooth and infeasible reference trajectories. The smooth trajectories are randomized degree-five polynomials and series of degree-five polynomials chained together. The infeasible trajectories are we refer to as *zigzag trajectories*, which are trajectories that linearly connect a series of random waypoints, and have either zero or undefined acceleration. The average speed of the infeasible trajectories is approximately $2 \, \text{m/s}$. See Appendix C for more details on the reference trajectories.

At the start of each episode, we apply a force perturbation $\boldsymbol{d}$ with randomized direction and strength in the range of $[-3.5 \, \text{m/s}^2, 3.5 \, \text{m/s}^2]$, representing translational disturbances. We then model time varying disturbance as Brownian motion; at each time step, we update $\boldsymbol{d} \leftarrow \boldsymbol{d} + \epsilon$, with $\epsilon \in \mathbb{R}^3$, $\epsilon \sim \mathcal{N}(\boldsymbol{0}, \boldsymbol{\Sigma} dt)$. We chose $\boldsymbol{\Sigma} = 0.01\boldsymbol{I}$. This is meant to model potentially complex time and state-dependent disturbances during inference time, while having few modeling parameters as we wish to demonstrate zero-shot generalization to complex target domains without prior knowledge. We run each episode for a total of 500 steps, corresponding to 10 seconds. By default, we set $H$ to $0.6 \, \text{s}$ with 10 feedforward reference terms. In Appendix A, we show ablation results for various different horizons.

We also note that stable training and best performance require fixing an initial trajectory for the first 2.5M steps of training (see Appendix A for more details). Only after that initial time period do we begin randomizing the trajectory. We train the policy using PPO for a total of 20M steps. Training takes slightly over 3 hours on an NVIDIA 3080 GPU.

## 4.2 Hardware Setup and the Low-level Attitude Rate Controller

We conduct hardware experiments with the Bitcraze Crazyflie 2.1 equipped with the longer $20 \, \text{mm}$ motors from the thrust upgrade bundle for more agility. The quadrotor as tested weighs $40 \, \text{g}$ and has a thrust-to-weight ratio of slightly under 2.

Position and velocity state estimation feedback is provided by the OptiTrack motion capture system at $50 \, \text{Hz}$ to an offboard computer that runs the controller. The Crazyflie quadrotor provides orientation estimates via a $2.4 \, \text{GHz}$ radio and control commands are sent to the quadrotor over the same radio at $50 \, \text{Hz}$. Communication with the drone is handled using the Crazyswarm API [34]. Body rate commands $\boldsymbol{\omega}_{\text{des}}$ received by the drone are converted to torque commands $\boldsymbol{\tau}$ using a custom low-level PI attitude rate controller on the firmware: $\boldsymbol{\tau} = -K_P^{\boldsymbol{\omega}}(\boldsymbol{\omega} - \boldsymbol{\omega}_{\text{des}}) - K_I^{\boldsymbol{\omega}} \int (\boldsymbol{\omega} - \boldsymbol{\omega}_{\text{des}})$. Finally, this torque command and the desired total thrust $f_{\Sigma,\text{des}}$ from the RL policy are converted to motor thrusts using the invertible actuation matrix.

## 4.3 Baselines

We compare our reinforcement learning approach against two nonlinear baselines: differential flatness-based feedback control and sampling-based Model Predictive Control (MPC) [4]. We also compare using $\mathcal{L}_1$ adaptive control, which we propose, against RMA.

**Nonlinear Tracking Controller and $\mathcal{L}_1$ Adaptive Control**  The differential flatness-based controller baseline consists of a PID position controller, which computes a desired acceleration vector, and a tilt-prioritized nonlinear attitude controller, which computes the body thrust $f_\Sigma$ and desired body angular velocity $\boldsymbol{\omega}_{\text{des}}$.

$$\boldsymbol{a}_{\text{fb}} = -K_P(\boldsymbol{p} - \boldsymbol{p}^d) - K_D(\boldsymbol{v} - \boldsymbol{v}^d) - K_I \int (\boldsymbol{p} - \boldsymbol{p}^d) + \boldsymbol{a}^d - \boldsymbol{g} - \hat{\boldsymbol{d}}/m, \tag{6a}$$

$$\boldsymbol{z}_{\text{fb}} = \frac{\boldsymbol{a}_{\text{fb}}}{||\boldsymbol{a}_{\text{fb}}||}, \quad \boldsymbol{z} = \boldsymbol{R}\boldsymbol{e}_3, \quad f_\Sigma = \boldsymbol{a}_{\text{fb}}^\top \boldsymbol{z} \tag{6b}$$

$$\boldsymbol{\omega}_{\text{des}} = -K_R \boldsymbol{z}_{\text{fb}} \times \boldsymbol{z} + \psi_{\text{fb}} \boldsymbol{z}, \quad \psi_{\text{fb}} = -K_{\text{yaw}}(\psi \ominus \psi_{\text{ref}}) \tag{6c}$$

where $\hat{\boldsymbol{d}}$ is the disturbance estimation. For the nonlinear baseline, we set $\hat{\boldsymbol{d}} = 0$, and for $\mathcal{L}_1$ adaptive control [11] we use (4) to compute $\hat{\boldsymbol{d}}$ in real time [11]. For our experiments, we set $K_P = \text{diag}([6 \; 6 \; 6])$, $K_I = \text{diag}([1.5 \; 1.5 \; 1.5])$, $K_D = \text{diag}([4 \; 4 \; 4])$, $K_R = \text{diag}([120 \; 120 \; 0])$, and $K_{\text{yaw}} = 13.75$. PID gains were empirically tuned on the hardware platform to track both smooth and infeasible trajectories while minimizing crashes.

**Nonlinear MPC and Adaptive Nonlinear MPC**  We use Model Predictive Path Integral (MPPI) [4] control as our second nonlinear baseline. MPPI is a sampling-based nonlinear optimal control

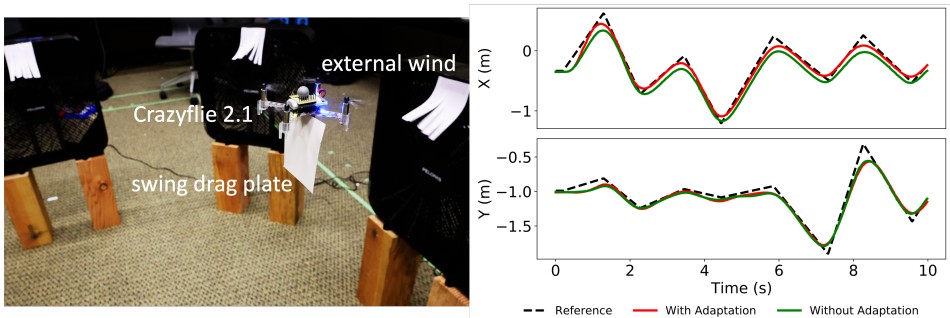

Figure 3: **Left**: Crazyflie 2.1 with a swinging cardboard drag plate in an unsteady wind field. **Right**: Comparison between our methods with and without adaptation with the drag plate on a zigzag trajectory. With wind added, adaptation is needed, otherwise the drone crashes.

technique that computes the optimal control sequence w.r.t. a known dynamics model and specified cost function. In our implementation, we use (1) ($\boldsymbol{d} = 0$) as the dynamics model with the body thrust $f_\Sigma$ and angular velocity $\boldsymbol{\omega}$ as the control input. The cost function is the sum of the position error norms along $k = 40$ horizon steps. We use 8192 samples, $dt = 0.02$, and a temperature of 0.05 for the softmax. For adaptive MPC, similar to prior works [18, 12], we augment the standard MPPI with the disturbance estimation $\hat{\boldsymbol{d}}$ from $\mathcal{L}_1$ adaptive control, which we refer to as $\mathcal{L}_1$-MPC.

**RMA**   We compare against RMA for our adaptive control baseline. Instead of using $\mathcal{L}_1$ to estimate $\hat{\boldsymbol{d}}$, we train an adaptation neural network $\psi$ that predicts $\hat{\boldsymbol{d}}$ from a history of state-action pairs using the RMA method (denoted DATT-RMA), similar to prior works [29]. We first train our policy $\pi$ in sim using PPO as usual, but conditioned on the ground truth $\boldsymbol{d}$. To train $\psi$, we then roll out $\pi$ with $\hat{\boldsymbol{d}}$ predicted by a randomly initialized $\psi$ for 500 timesteps. $\psi$ is then trained with supervised learning in order to minimize the loss $\|\hat{\boldsymbol{d}} - \boldsymbol{d}\|$. We repeat this process for 10000 iterations, when the loss converges. Our adaptation network $\psi$ takes as input the previous seen 50 state-action pairs, and the architecture consists of 3 1D convolutional layers with 64 channels and a kernel size of 8 for each, followed by 3 fully connected layers of size 32 and ReLU activations.

## 4.4   Arbitrary Trajectory Tracking

We first evaluate the trajectory tracking performance of DATT compared to the baselines in the absence of disturbances. We test on both infeasible zigzag trajectories and smooth polynomial trajectories. Each controller is run 2 times on the same bank of 10 random zigzag trajectories and 10 random polynomials. Results are shown in Table 1. For completeness, we also compare with the tracking performance of adaptive controllers in the absence of any disturbances. We also compare our method to a version without adaptation, meaning that we enforce $\hat{\boldsymbol{d}} = \boldsymbol{0}$.

| Arbitrary trajectory tracking without external disturbances | | | |
|---|---|---|---|
| Method | Smooth trajectory | Infeasible trajectory | Inference time (ms) |
| Nonlinear tracking control | $0.098 \pm 0.012$ | *crash* | 0.21 |
| $\mathcal{L}_1$ adaptive control | $0.091 \pm 0.009$ | *crash* | 0.93 |
| MPC | $0.104 \pm 0.009$ | $0.183 \pm 0.027$ | 12.62 |
| $\mathcal{L}_1$-MPC | $0.088 \pm 0.010$ | $0.181 \pm 0.031$ | 13.10 |
| DATT (w/ $\hat{\boldsymbol{d}} = 0$) | $\mathbf{0.054} \pm 0.013$ | $\mathbf{0.089} \pm 0.026$ | 2.41 |
| DATT | $\mathbf{0.049} \pm 0.017$ | $\mathbf{0.083} \pm 0.023$ | 3.17 |

Table 1:  Tracking error (in m) of DATT vs. baselines, without any environmental disturbances (no wind or plate). *crash* indicates a crash for all ten trajectory seeds.

We see that DATT achieves the most accurate tracking, with a fraction of the compute cost of MPC. With our current gains, the nonlinear and $\mathcal{L}_1$ adaptive control baselines are unable to track the infeasible trajectory. With reduced controller gains, it is possible these controllers would not crash when tracking the infeasible trajectories, but doing so would greatly decrease their performance for smooth trajectories.

## 4.5 Adaptation Performance in Unknown Wind Fields with a Drag Plate

To evaluate the ability of DATT to compensate for unknown disturbances, we test the Crazyflie in a high wind scenario with three fans and an attached soft cardboard plate hanging below the vehicle body. Figure 3 shows this experimental setup. We note that this setup differs significantly from simulation — the placement of the fans and the soft cardboard plate creates highly dynamic and state dependent force disturbances, as well as torque disturbances, yet in simulation we model only the force disturbance as a simple random walk. However, our policy is able to generalize well zero-shot to this domain, as shown in Table 2.

| | Arbitrary trajectory tracking with external disturbances | | | |
|---|---|---|---|---|
| Method | Smooth traj. w/ plate | Smooth traj. w/ plate & wind | Infeasible traj. w/ plate | Infeasible traj. w/ plate & wind |
| $\mathcal{L}_1$ adaptive control | $0.163 \pm 0.013$ | $0.184 \pm 0.020$ | *crash* | *crash* |
| $\mathcal{L}_1$-MPC | $0.121 \pm 0.010$ | $0.181 \pm 0.04$ | $0.216 \pm 0.028$ | $0.243 \pm 0.026$ |
| DATT (w/ $\hat{d} = 0$) | $0.091 \pm 0.040$ | $0.118 \pm 0.054$ | $0.143 \pm 0.031$ | *crash* |
| DATT-RMA | $0.091 \pm 0.049$ | $0.115 \pm 0.071$ | $0.164 \pm 0.051$ | $0.193 \pm 0.075$ |
| DATT | $\mathbf{0.063} \pm 0.052$ | $\mathbf{0.095} \pm 0.053$ | $\mathbf{0.122} \pm 0.041$ | $\mathbf{0.161} \pm 0.056$ |

Table 2: Tracking error (in m) of DATT vs. baselines, with an attached plate and/or wind. Results are effectively for zero-shot generalization, as we do not model a plate, torque disturbances, or exact force disturbances in simulation.

In Table 2, we see that the baseline nonlinear adaptive controller is unable to track infeasible trajectories, similar to the experiment without adaptation. Our method with adaptation enabled is able to track all the trajectories tested, with the lowest tracking error. We also verify that using $\mathcal{L}_1$ adaptive control results in better performance than using RMA. We note that this is due to a large sim2real gap with the adaptation network for RMA, which we discuss in the Appendix. Figure 3 shows the difference in tracking performance between our method using adaptive control and our method without, on an example zigzag trajectory with a drag plate. We see that our approach of integrating $\mathcal{L}_1$ adaptive control with our policy controller is effective in correcting the error introduced by the presence of the turbulent wind field and plate. Our method performs better than $\mathcal{L}_1$-MPC without any knowledge of the target domain, and with a fraction of the compute cost. Figures 5 and 6 in the Appendix visualizes the tracking performance of DATT vs. $\mathcal{L}_1$-MPC on a infeasible and smooth trajectory, respectively.

## 5 Limitations and Future Work

Our choice of hardware presents some inherent limitations. The relatively low thrust-to-weight ratio of the Crazyflie (less than 2) means that we are unable to fly very agile or aggressive trajectories on the real drone or perform complex maneuvers such as a drone flip mid-trajectory. For this reason, we focused on $xy$-planar trajectories in this paper, and did not vary the $z$ direction. However, our method provides the framework for performing accurate tracking for any trajectory, as we note we are able to perform a much larger range of agile maneuvers in simulation, including flips.

Our simulator is only an approximation of the true dynamics. For example, we model the lower-level angular velocity controller with a simplified first-order time delay model, which limits sim2real generalization for very agile tasks. Furthermore, our force disturbance model is highly simplified in sim, which only approximates the highly time- and state-dependent force and torque disturbances the drone can encounter in reality. However, we show that we can already achieve good zero-shot generalization to a highly dynamic environment and challenging tasks.

We also note that our training process has fairly high variance and can be sensitive to the hyperparameters of the PPO algorithm, typical of RL. As seen in Appendix A, we use a few tricks for stable learning, including fixing the reference trajectory for the first 2.5M training steps. Future work is needed to understand the role of these architectural and training features and help inform the best algorithm design and training setup.

**Acknowledgments**

We would like to acknowledge the Robot Learning Lab at the University of Washington for providing the resources for this paper. We would also like to thank the reviewers for their helpful and insightful comments.

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

# A Ablations

| Ablation | Tracking error (sim) (m) |
|---|---|
| No body frame | *failed* |
| No fixed intial reference | $0.437 \pm 0.08$ |
| No feedback term | $0.077 \pm 0.011$ |
| Feedforward horizon 1 ($H = 0.02s$) | *failed* |
| Feedforward horizon 5 ($H = 0.3s$) | $0.240 \pm 0.008$ |
| Feedforward horizon 10 ($H = 0.6s$) (used in main experiments) | $0.055 \pm 0.007$ |
| Feedforward horizon 15 ($H = 0.9s$) | $0.073 \pm 0.010$ |
| Feedforward horizon 20 ($H = 1.2s$) | $0.101 \pm 0.018$ |
| Base policy (no ablation) | $0.046$ |

Table 3: Tracking error (in m), in simulation, of various ablations after 15M training steps. *Failed* indicates the drone diverges from the reference trajectory. Tracking error is with respect to infeasible zigzag trajectories. The ablations are done without adaptation, and with no disturbances in the environment. 5 runs were attempted for each ablation.

We test various ablations of our primary method, with results shown in Table 3. In particular, we test

- **No body frame**: With our training setup, we found that transforming all state inputs (except for the orientation) into the body frame was necessary for accurate trajectory tracking. This ablation tests our method, but with the position $p$, velocity $v$, and reference positions in the world frame instead of the body frame.

- **No fixed initial reference** This ablation removes the initial 2.5M training steps where we do not randomize the reference trajectory. We see that PPO converges to a much worse tracking performance. We note that the choice of the initial fixed reference does not have much impact on the variance of training, only the existence of the fixed reference.

- **No feedback term** We remove the feedback term $\boldsymbol{R}^\top(\boldsymbol{p}_t - \boldsymbol{p}_t^d)$ from our controller inputs. This term might appear redundant with the reference trajectory, but we find explicitly conditioning on the feedback error consistently results in slightly more accurate tracking.

- **Feedforward horizon** We test varying sizes of our feedforward horizon. In Table 3, Feedforward horizon $N$ refers to passing in $N$ future reference positions. As described in Section 3.2, we linearly space the $N$ reference positions across time from $t$ to $t + H$.

- **Base policy** For comparison, we list the tracking error in sim of the main policy that we use in our experiments section.

**Adaptive Control in Simulation** As seen in Table 4, in simulation, using RMA as the adaptive control strategy actually yields slightly better performance than $\mathcal{L}_1$ adaptive control. However, on the real drone, as we report in Table 2, RMA performs significantly worse than $\mathcal{L}_1$ adaptive control, indicating a significant sim2real gap. This is likely because the adaptation network in DATT-RMA is highly susceptible to the domain shift in state-action pair inputs on the real drone, while the closed-loop nature of L1 guarantees fast disturbance estimation for any state-action pairs.

| Method | Tracking error (sim) (m) |
|---|---|
| DATT (w/ disturbances) | $0.062 \pm 0.011$ |
| DATT-RMA (w/ disturbances) | $0.055 \pm 0.009$ |

Table 4: Tracking error (in m), in simulation, of standard DATT (using $\mathcal{L}_1$ adaptive control) and DATT-RMA with random force disturbances

## B    Training Details and Network Architecture

Training is done with the PPO implementation in the Stable Baselines3 library [35]. All PPO parameters are left as default.

The feedforward encoder architecture consists of 3 1-D convolution layers with ReLU activations that project the reference positions into a 32-dim representation for input to the main policy. Each 1-D convolution has 16 filters with a kernel size of 3. The main policy network is a 3-layer MLP with 64 neurons per layer and ReLU activations, and the value network shares this structure.

## C    Reference Trajectory Details

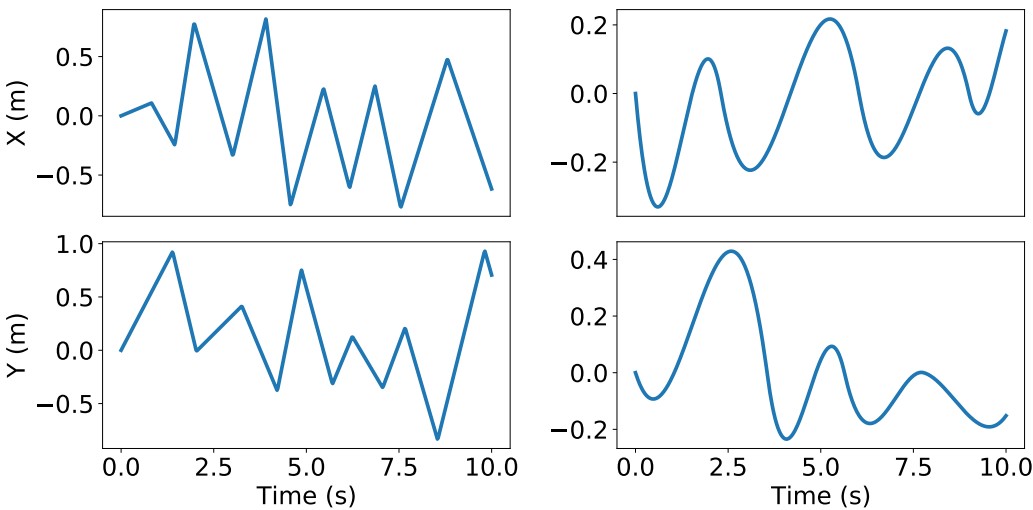

Figure 4: **Left:** Example of a random zigzag trajectory (infeasible). **Right:** Example of a random chained polynomial trajectory (smooth).

### C.1    Smooth Trajectory

For smooth trajectories, we include a mix of degree 5 polynomials and *chained polynomials*. Polynomials start at $x = 0$ and $y = 0$, and return to the origin after $10\,\mathrm{s}$, corresponding to our episode length. They are randomly generated by randomly selecting initial and end conditions. Chained polynomials are a series of random polynomials. We generate these trajectories by randomly selecting "nodes" at $x = 0$ and $y = 0$ at random times between $0\,\mathrm{s}$ and $10\,\mathrm{s}$, and fitting degree 5 polynomials between each node, ensuring that first, second, and third order derivatives are continuous at each node. Note that these trajectories are not guaranteed to be feasible, although in practice they are easy to track as they are highly smooth.

### C.2    Infeasible Trajectory

We use a class of what we refer to as *zigzag trajectories*. We generate these trajectories by randomly selecting time intervals between $0.5$ and $1.5$ seconds, randomly generating waypoints after each time interval, and linearly connecting each waypoint. The waypoints can vary from $-1\,\mathrm{m}$ to $1\,\mathrm{m}$ in both the $x$ and $y$ directions. By training on these zigzags, we are able to generalize well to a wide variety of trajectories, including polygons and stars as seen in Figure 1, which are similar to random zigzags.

### C.3    Additional Figures of Results

We show additional figures from our results from Table 2. Figure 7 shows the values of the predicted $\hat{d}$ over time on an environment with wind versus one without wind. Figure 5 and Figure 6 show our tracking performance against $\mathcal{L}_1$-MPC for a smooth and infeasible trajectory, respectively.

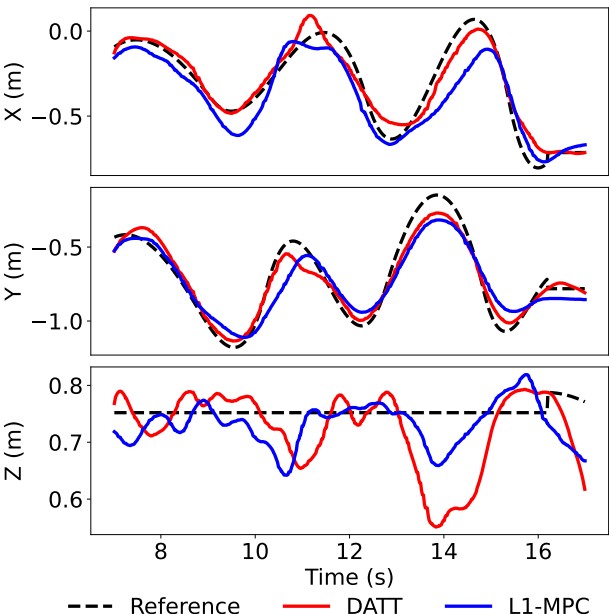

Figure 5: Performance of DATT against $\mathcal{L}_1$-MPC on a smooth trajectory with both wind and a plate attached.

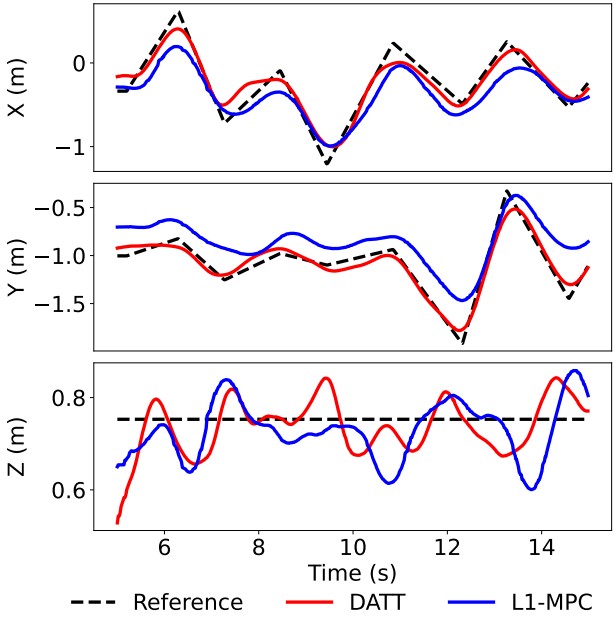

Figure 6: Performance of DATT against $\mathcal{L}_1$-MPC on an infeasible trajectory with both wind and a plate attached.

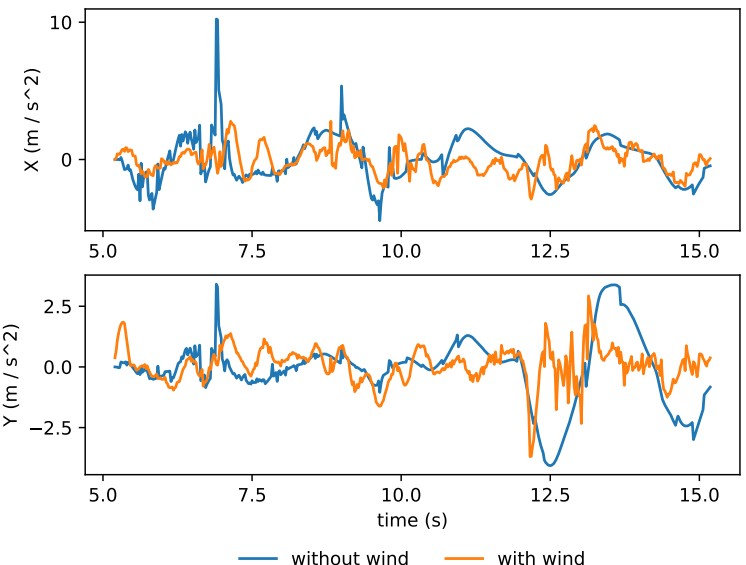

Figure 7: Predicted $\hat{d}$ terms on two infeasible trajectories, one with wind, one without wind but with an air drag plate.

