# OpenReview forum: "DATT: Deep Adaptive Trajectory Tracking for Quadrotor Control"
_robot-learning.org/CoRL/2023/Conference — CoRL 2023 Oral_

### Official Review · Reviewer_fQDU · 2023-07-18

**Confidence:** 5
**Originality:** Very Good
**Technical Quality:** Good
**Clarity Of Presentation:** Excellent
**Impact:** 4

**Recommendation:**

Strong Accept: I recommend accepting the paper and will argue for my recommendation even if other reviewers hold a different opinion.

**Review:**

Below are my reviews of this paper

0. The paper is written well. Many questions while reading the paper are answered in the supplementary material. The experimental validations look good.

1. This is an interesting paper that tries to build the latent space based on the position error to the desired future positions, which should carry useful information on whether the desired positions to follow next will be feasible. The idea of training with simulated (known) disturbance and deploying an online estimator of the disturbance for the actual system is not new: the Rapid Motor Adaptation (RMA) mechanism (reference [26]) has proposed a similar idea, and it worked well for quadrupedal and quadrotors. The authors should know that RMA has been applied to quadrotors [R1] and must compare the difference in methodology and experimental verifications.

2. The authors only compare the proposed method's performance with conventional non-learning controllers. However, given that neural networks (NNs) contain way more parameters than conventionally designed controllers, such a comparison can be an apple-to-orange comparison, which for sure the NN-parameterized controller will beat the conventional ones given the former's larger capabilities with more parameters. This is another reason why a comparison to the RMA [R1] should be conducted.

3. When referring to the L1 adaptive control literature, the authors may want to cite the guaranteed bounded disturbance error using the state predictor with the piecewise constant adaptation law, e.g., proposition 4 in [11] and lemma 3 in [R2].

4. In the experiments, it's fundamental to add unknown weights as a benchmark comparison (unknown mass induces unknown forces when the quadrotor tries to generate forces in the horizontal directions). Please consider adding this comparison.

5. In Figs. 5 and 6 in the appendix, the authors didn't show the tracking performance of the baseline L1 adaptive control method (only L1-MPC was shown), which was mentioned in Table 2. Please add the corresponding results.

Reference
[R1] Zhang, Dingqi, Antonio Loquercio, Xiangyu Wu, Ashish Kumar, Jitendra Malik, and Mark W. Mueller. "Learning a Single Near-hover Position Controller for Vastly Different Quadcopters." In 2023 IEEE International Conference on Robotics and Automation (ICRA), pp. 1263-1269. IEEE, 2023.
[R2] Zhao, Pan, Yanbing Mao, Chuyuan Tao, Naira Hovakimyan, and Xiaofeng Wang. "Adaptive robust quadratic programs using control Lyapunov and barrier functions." In 2020 59th IEEE Conference on Decision and Control (CDC), pp. 3353-3358. IEEE, 2020.

**Quality Of The Limitations Section:**

Limitations are addressed clearly

**Questions For Rebuttal:**

1. It's not clear why the position and velocity have to be converted to the body frame as inputs to the policy /pi. Why cannot those in the inertial frame work here?

2. It seems a limitation where the training is conducted with constant force perturbation. Why not use a time-varying disturbance? Wouldn't the policy perform better when you train it with more realistic data?

3. What about generalizing to different quadrotor platforms? The experiments are only demonstrated on a crazyflie quadrotor.

4. In Section 5, the statement "although our method also improves the performance in the z direction (Fig. 1)" seems to contradict the results shown in Figs. 5 and 6, where the tracking on the z-axis is not as good as the L1-MPC. Please either fix the statement or provide explanations.

updated on Aug. 18: my concerns have been addressed by the authors. I have upgraded my score to strongly accept.

**Robotics Focus:**

Sufficient demonstration on hardware

**Summary Of Paper:**

This paper proposes a control architecture for a quadrotor to track arbitrary (possibly non-feasible) trajectories subject to uncertainties using reinforcement learning. Two components are learned: one is a feedforward embedding h, which takes the inputs of the position error to future planned positions on the trajectory and projects it to a latent space; the other one is the feedback policy that takes in the state information, latent-space represented terms from the embedding h, and disturbance and outputs the control actions. Both the two components are trained via reinforcement learning in simulations. The online deployment uses the uncertainty estimation obtained via the L1 adaptive control. Experimental validations are provided to demonstrate the performance of the proposed architecture.

**Summary Of Recommendation:**

The recommendation is made based on the comments listed above. Please provide extra experimental results and supply more details to the questions above.

updated on Aug. 18: the reviewer's questions have been addressed.

---

> ### Author Response · Authors · 2023-08-12
> **Response to Reviewer fQDU**
>
> Thank you for your detailed feedback. We address your comments below:
>
> > The authors should know that RMA has been applied to quadrotors [R1] and must compare the difference in methodology and experimental verifications.
>
> We note that we have cited the paper [R1] as [23] in our paper, although the paper seems to have changed its name from when our citation was originally made. We will update the citation in our revision.
>
> As mentioned in our general response, we have added a comparison of DATT-L1 vs. DATT-RMA, finding that DATT-RMA suffers from a greater sim2real gap and ultimately worse performance. We note that our problem setting is quite different than [23], which might explain the difference in sim2real behavior. [23] tests large single domain shifts (i.e. adding a payload mid-flight) for hover only, while we aim to design an adaptive trajectory tracking controller for arbitrary force disturbances. We do believe that learning based adaptive controllers are a promising direction, and believe more carefully designed learning strategies can be developed with greater success. Our DATT framework easily allows extension to any such future method.
>
> > The authors only compare the proposed method's performance with conventional non-learning controllers. However, given that neural networks (NNs) contain way more parameters ...
>
> We note that naive implementations of training neural network controllers with PPO perform worse than classical controllers [A1], and we believe that so far, it has not been firmly established that learning based controllers are the way to go for arbitrary trajectory tracking with adaptation.
>
> As per the general response, we have included a comparison to DATT-RMA.
>
> [A1] N. Wiedemann, V.  Wüest, A. Loquercio, M. Müller, D. Floreano, D. Scaramuzza.  Training Efficient Controllers via Analytic Policy Gradient. arXiv preprint arXiv:2209.13052, 2023
>
> > When referring to the L1 adaptive control literature, the authors may want to cite the guaranteed bounded disturbance error ...
>
> Thank you for the recommendation, we will add these references to our final submission. We do want to point out that DATT is intended to be a general recipe. In this paper, we only use the basic variant of L1, but in future, we would like to explore more advanced variants such as [R2].
>
> > In the experiments, it's fundamental to add unknown weights as a benchmark comparison (unknown mass induces unknown forces when the quadrotor tries to generate forces in the horizontal directions). Please consider adding this comparison.
>
> The swinging paper drag plate we test with weighs approximately 3 grams, which constitutes about 7.3% of the mass of the Crazyflie without the plate. Considering the relatively low thrust-to-weight ratio of the Crazyflie, the weight is significant, and acts as a payload. The mass of the plate is not known to the policy controller, and the plate induces unknown forces as described, so our experiments with a plate and no wind doubles as an unknown baseline. Furthermore, in combination with the fans, the plate will generate significant unknown time- and state-dependent disturbances in all directions (including the vertical direction), which should be significantly more challenging than a payload, and capture almost all the forces that a payload could induce, as well as inducing significant torque disturbances.
>
> > In Figs. 5 and 6 in the appendix, the authors didn't show the tracking performance of the baseline L1 adaptive control method (only L1-MPC was shown), which was mentioned in Table 2. Please add the corresponding results.
>
> We will update these figures.
>
> > It's not clear why the position and velocity have to be converted to the body frame as inputs to the policy /pi.
>
> As shown in our ablations, we found that body frame inputs were critical to the success of our controller, although we are not aware of a simple theoretical explanation. We guess it is related to the specific optimization landscape and RL algorithm associated with the problem, and we believe this is an interesting area for further investigation.
>
> > It seems a limitation where the training is conducted with constant force perturbation. Why not use a time-varying disturbance?
>
> Please see the general response.
>
> > What about generalizing to different quadrotor platforms?
>
> The main goal of our paper is to show generalization with respect to arbitrary trajectories and disturbances, but our simulation is relatively platform agnostic, as we assume normalized mass and inertia, and output mass normalized thrust. Generalizing to a different drone would require measuring the mass of the new drone, as well as potentially tuning the low-level controller in simulation.
>
> > In Section 5, the statement "although our method also improves the performance in the z direction (Fig. 1)" seems to contradict the results shown in Figs. 5 and 6, ...
>
> Thank you for bringing this to our attention, we will reword this statement.

---

> > ### Comment · Reviewer_fQDU · 2023-08-15
> > **my concerns are addressed**
> >
> > Thank you for addressing my concerns. Minor comments are made after the "general response."

---

### Official Review · Reviewer_Tsi5 · 2023-07-19

**Confidence:** 4
**Originality:** Good
**Technical Quality:** Very Good
**Clarity Of Presentation:** Good
**Impact:** 3

**Recommendation:**

Weak Accept: I recommend accepting the paper, but will not argue for my recommendation if the majority of other reviewers have a different opinion.

**Review:**

Combining off-line learning and online adaptation (in a feedforward-feedback sense) seems a very promising technique for the control of dynamical systems.
The concept is not entirely new, a recent work introduced a very similar concept with feedforward-feedback adaptive control:
Verginis et al, 'Non-parametric neuro-adaptive control', ECC 2023, https://ieeexplore.ieee.org/document/10178288
and an association/comparison with it should be made.

The paper is generally well-written, and the algorithm is verified extensively in simulation and experiments.
I believe it mostly needs clarification in Section 3 regarding the proposed methodology, which is not completely clear.
More specific comments follow below.


**Quality Of The Limitations Section:**

Limitations are addressed clearly

**Questions For Rebuttal:**

- The recent work 'Non-parametric neuro-adaptive control' should be referenced and a comparison (in the related work section) should be made.
- The disturbance d is time-varying and as far as I understand the training uses a constant one. Have you checked training with a time-varying one? Do you expect it would yield better results?
- It is not clear what the embedding h represents. What does 'encodes the desired reference trajectory' mean? It is stated that
'Thus, we input just the desired positions into a feed-forward encoder φ, which learns the feedforward embedding that contains the information of the desired future reference positions' but in (2a), the inputs are the errors. Is it a feedback-control term learned with a neural network?
- Where does (4) come from? I understand the authors refer to an arxiv paper, but it would be very beneficial to briefly explain what the intuition of (4) and how it is derived.

**Robotics Focus:**

Sufficient demonstration on hardware

**Summary Of Paper:**

The paper proposes an algorithm for controlling a quadrotor to track arbitrary trajectories. Deep RL is used along with disturbance adaptation in real-time.  With respect to previous works, the algorithm can track infeasible trajectories (in the sense of non-differentiable ones) with better accuracy. Extensive simulation and experimental results are presented.

**Summary Of Recommendation:**

I believe there is theoretical novelty in the paper and good experimental demonstrations. However, the paper needs to be revised, mostly on its presentation in the sense of clarification of the algorithm.

---

> ### Author Response · Authors · 2023-08-12
> **Response to Reviewer Tsi5**
>
> Thank you for your comments and feedback. We have addressed your questions below:
>
> > The recent work 'Non-parametric neuro-adaptive control' should be referenced and a comparison (in the related work section) should be made.
>
> Thank you for bringing this paper to our attention, although we would like to note this paper appears to have been published after the submission deadline for CoRL. As briefly discussed in our general response, we would like our method to provide a general framework for learning trajectory tracking with RL in drones and have designed it in such a way to be easily extensible to other adaptation strategies. We believe this paper is relevant, and we will edit our final submission to include a citation.
>
> > The disturbance d is time-varying and as far as I understand the training uses a constant one. Have you checked training with a time-varying one? Do you expect it would yield better results?
>
> Please see the general response.
>
> > It is not clear what the embedding h represents. What does 'encodes the desired reference trajectory' mean? It is stated that 'Thus, we input just the desired positions into a feed-forward encoder φ, which learns the feedforward embedding that contains the information of the desired future reference positions' but in (2a), the inputs are the errors. Is it a feedback-control term learned with a neural network?
>
> We would like to clarify that $\phi$ in essence is simply more DNN layers jointly trained as part of the entire neural network policy controller. We use $h$ to more easily notationally represent the lower dimensional encoding of the feedforward trajectory after passing through the network layers in $\phi$. Intuitively, $h$ should contain all the information of the desired future positions in a lower dimensional representation.The inclusion of such layers in $\phi$ to process large time series data is common, for example in [26] and in [7].
> 	As for the input to the network, the sentence referenced should read 'Thus, we input just the desired positions into a feed-forward encoder φ, in the body-frame, [...]`, which is consistent with (2a). To clarify, these terms allow us to do feedforward control, not feedback control. We will update the wording in our final submission.
>
> > Where does (4) come from? I understand the authors refer to an arxiv paper, but it would be very beneficial to briefly explain what the intuition of (4) and how it is derived.
>
> Equation (4) states the standard L1 adaptive control law, originally from [A] and adopted in [11]. Generally speaking, L1 adaptive control first builds a closed-loop estimator to compute the difference between predictive and true disturbance, and then leverage a low pass filter to update the prediction. The gains of such estimator and filter are carefully designed to guarantee stability and robustness. We will add additional intuition to our final submission.
>
> [A] N. Hovakimyan and C. Cao. 1 Adaptive Control Theory: Guaranteed Robustness with Fast Adaptation. Society for Industrial and Applied Mathematics, 2010.

---

> > ### Comment · Reviewer_Tsi5 · 2023-08-13
> > **Thank you note**
> >
> > I thank the authors for the reply, my comments have been addressed.

---

### Official Review · Reviewer_Bvch · 2023-07-19

**Confidence:** 5
**Originality:** Very Good
**Technical Quality:** Very Good
**Clarity Of Presentation:** Very Good
**Impact:** 4

**Recommendation:**

Strong Accept: I recommend accepting the paper and will argue for my recommendation even if other reviewers hold a different opinion.

**Review:**

The work is really nice and clearly presented. No apparent weaknesses.

**Quality Of The Limitations Section:**

Limitations are addressed clearly

**Questions For Rebuttal:**

I have no questions for rebuttal and recommend the paper be published as submitted.

**Robotics Focus:**

Sufficient demonstration on hardware

**Summary Of Paper:**

This paper is about precision trajectory tracking for quadrotors. It presents a learning-based approach that can track challenging trajectories in the presence of large disturbances. After RL training in simulation, the approach transfers to the real world. The experimental results are impressive.

**Summary Of Recommendation:**

This a very nice paper and a pleasure to read. The algorithm is well explained, the results are convincing, and the demonstration on hardware is excellent.

---

> ### Author Response · Authors · 2023-08-12
> **Response to Reviewer Bvch**
>
> Thank you for the positive review. We would like to make you aware of our general response, which adds additional context and experiments that we will incorporate into our final submission.

---

> > ### Comment · Reviewer_Bvch · 2023-08-15
> > **Thanks !**
> >
> > Thanks to the authors for their response (both to this review and others).

---

### Official Review · Reviewer_cDww · 2023-07-23

**Confidence:** 4
**Originality:** Good
**Technical Quality:** Very Good
**Clarity Of Presentation:** Very Good
**Impact:** 3

**Recommendation:**

Weak Accept: I recommend accepting the paper, but will not argue for my recommendation if the majority of other reviewers have a different opinion.

**Review:**

Strengths
=========
- The paper is clearly written, reads well, and is easy to follow.
- The experiments are performed on a physical robot.
- The paper is very clear and transparent about the limitations of the presented work.


Constructive Criticism
==================
- L16: UAV = *unmanned* aerial vehicle, not uninhabited aerial vehicle
- In the simulation, the proposed method is conditioned on ground-truth translational disturbance, while in the real world, the translational disturbance is estimated by an L1 adaptive controller. Is there a reason to not run the same L1 adaptive controller in simulation as well? Would doing so potentially reduce the sim2real gap?
- The proposed controller is able to reject external disturbances based on the information contained in the observed positional tracking error and the estimated force disturbance generated from the L1 adaptive controller. Previous work (ref [28] in the paper) proposed to use a history of state-action pair observations instead to achieve a similar effect. Would the proposed controller potentially benefit from such a history-based observation as well?
- Minor in Figure 1: always add a grid to every plot, it improves the reading experience.
- Minor: how does the performance of MPPI change with varying sample numbers?
- Minor thought to think about, does not require to be addressed: how much sense does it make to evaluate tracking performance on infeasible trajectories? Isn't such a metric ill-posed in the first place? Evaluating tracking performance on infeasible trajectories becomes extremely sensitive to controller tuning.

**Quality Of The Limitations Section:**

Limitations are addressed clearly

**Questions For Rebuttal:**

In the simulation, the proposed method is conditioned on ground-truth translational disturbance, while in the real world, the translational disturbance is estimated by an L1 adaptive controller. Is there a reason to not run the same L1 adaptive controller in simulation as well? Would doing so potentially reduce the sim2real gap?




**Robotics Focus:**

Sufficient demonstration on hardware

**Summary Of Paper:**

The paper proposes a learning-based controller for quadrotors that is robust to external disturbances such as unsteady wind fields. The learned controller is trained via model-free reinforcement learning in simulation, where it is exposed to random force disturbances while being tasked to track random trajectories. After training the controller for 2e7 environment steps, the controller is deployed zero-shot on a physical quadrotor.

The presented method is compared against a differential flatness-based controller and a sampling-based model predictive controller on the task of trajectory tracking with and without external wind disturbances. The proposed learning-based controller outperforms both baselines.

**Summary Of Recommendation:**

My recommendation is based on the good presentation of the paper, the convincing set of baselines that the presented approach is evaluated against, and the reported performance of the method.

---

> ### Author Response · Authors · 2023-08-12
> **Response to Reviewer cDww**
>
> Thank you for reviewing our work! We address your questions and feedback below:
>
> > unmanned aerial vehicle, not uninhabited aerial vehicle
>
> We have corrected the term.
>
> > In the simulation, the proposed method is conditioned on ground-truth translational disturbance, while in the real world, the translational disturbance is estimated by an L1 adaptive controller. Is there a reason to not run the same L1 adaptive controller in simulation as well? Would doing so potentially reduce the sim2real gap?
>
> Please see our General Response.
>
> > The proposed controller is able to reject external disturbances based on the information contained in the observed positional tracking error and the estimated force disturbance generated from the L1 adaptive controller. Previous work (ref [28] in the paper) proposed to use a history of state-action pair observations instead to achieve a similar effect. Would the proposed controller potentially benefit from such a history-based observation as well?
>
> As discussed in the General Response, using a history of state-action pairs for adaptation is the principal idea behind RMA [26]. In theory, a neural network could leverage this additional information for more accurate control, but as discussed in the general response, we encountered significant challenges with sim2real, resulting in worse performance than L1. As alluded to in [28], another option for leveraging state-action histories is an “end-to-end” neural network approach, where the entire policy is conditioned on a history of state-action pairs, and any disturbance d is implicitly learned from the history. However, we found that this method was not able to train well with PPO in sim. It is likely that training such a method would require either more extensive tuning of PPO parameters, or another RL algorithm.
> Furthermore, we would expect this method to also have sim2real issues with domain shift of the history, as RMA does. Investigating novel methods of training these types of controllers is an interesting direction for future work.
>
> We would like to point out that an L1 controller does implicitly leverage history information, as the low pass filter is implemented as an exponential moving average of a history of d_hat estimations. However, L1 adaptive control is essentially closed-loop estimation, and is not significantly affected by out of distribution state-action pairs, meaning it should suffer less from sim2real.
>
> > Minor in Figure 1: always add a grid to every plot, it improves the reading experience.
>
> We will update the figure.
>
> > Minor thought to think about, does not require to be addressed: how much sense does it make to evaluate tracking performance on infeasible trajectories? Isn't such a metric ill-posed in the first place? Evaluating tracking performance on infeasible trajectories becomes extremely sensitive to controller tuning.
>
> This is an interesting philosophical question. While tracking infeasible trajectories exactly is indeed impossible, we believe that doing the best we can given the limitations of the system is a useful problem, given that minimizing L2 norm error (or any other user-specified reward function) to a set of position waypoints is valuable to solve in practice. Controller tuning is a tricky issue, but it is very hard to increase performance just by increasing gains, for example, because stability becomes a major issue.

---

> > ### Comment · Reviewer_cDww · 2023-08-16
> > **Response to authors**
> >
> > Thank you for your detailed response. I don't have any further questions.

---

### Author Response · Authors · 2023-08-12
**General Response - Clarifications on Training Setup**

We would like to thank all reviewers for their helpful comments. We wanted to provide insight into our choice of experimental setup, as well as present additional experiments based on the feedback.

Reviewers raised points about how we handled the disturbance in simulation. Our choice to use a constant disturbance across each training episode in simulation resulted from our goal to provide a domain-agnostic framework for training adaptive neural network controllers for trajectory tracking using RL. Because we do not know the time and state dependence of the disturbances in the target domain, it is not possible to exactly model the disturbance during training. In fact, carefully tuning the disturbance model in sim could risk overfitting. A randomized constant disturbance requires very few modeling parameters, and thus was a natural choice to demonstrate zero-shot generalization to complex target domains. Of course, with prior information about a target domain (e.g., an accurate air drag model), a better controller could be trained.

In a similar vein, DATT is not specific to L1-adaptive control. While we tested with L1 adaptive control, our goal was to provide a modular framework where the specific method for adaptation can be chosen after training the policy . This is why we chose to condition our policy controller on the ground truth disturbance d in sim and not explicitly the L1 output: this means that any adaptive control method that is able to estimate d online can be used with the same trained controller.

For example, one alternative adaptation method is RMA [26], which uses a neural network to predict d based on a history of state-action pairs. Our current training method allows easy swapping of the L1 adaptation with an adaptation network, which can be trained using the same main policy controller. As suggested by reviewers, we have added the comparison of DATT with L1 (DATT-L1) and DATT with RMA instead of with L1 for adaptation (DATT-RMA).

---

> ### Author Response · Authors · 2023-08-12
> **General Response - New Experiments**
>
> While we aim to demonstrate a general strategy for training trajectory tracking controllers usig RL with zero-shot generalization, we do believe that tuning our training procedure to a known desired domain and adaptation strategy would yield better results. For completeness, we wanted to provide the additional experimental results in which we used the L1-controller in simulation alongside time varying disturbance, as suggested by reviewers. We will also adopt these new results in the final version.
>
> We model our time varying disturbance as Brownian motion, i.e. at each time step, we update $d \leftarrow d + \epsilon$, $\epsilon \sim \mathcal{N}(0, \sigma^2 dt)$. We choose this model as it is relatively light on parameters, but could potentially better capture the complex dynamics of the fan and plate system, which has shifting disturbances.
>
> We also tested running the L1 controller in sim rather than conditioning on the ground truth disturbance.
>
> All other training configurations are identical as described in the paper (2 runs each for 10 randomized trajectories for each category). Results are shown in the table below.
>
> | Method | Smooth traj. w/ plate | Smooth traj. w/ plate & wind | Infeasible traj. w/ plate | Infeasible traj. w/ plate & wind |
> | - | - | - | - | - |
> | DATT (L1 in sim) + L1 |  $0.090 \pm 0.031$ | $0.126 \pm 0.045$ | $0.148 \pm 0.062$ | $0.175 \pm 0.070$ |
> | DATT (time varying d in sim)  + L1 | $0.091 \pm 0.042$ | $0.107 \pm 0.055$ | $0.152 \pm 0.068$ | $0.178 \pm 0.076$ |
> | DATT (L1 + time varying d in sim) + L1 | $0.063 \pm 0.052$ | $0.095 \pm 0.053$ | $0.122 \pm 0.041$ | $0.161 \pm 0.056$ |
> | DATT (time varying d in sim) + RMA | $0.091 \pm 0.049$ | $0.115 \pm 0.071$ | $0.164 \pm 0.051$ | $0.193 \pm 0.075$ |
>
> With either L1 and time varying disturbance in sim, our controller indeed improves in performance compared to our original controller. Using both together further improves performance, and results in our best performance, which is to be expected as the simulation is more accurate to our tested domain.
>
> However, we note that this choice of disturbance model introduces an extra parameter $\sigma$ (which we tuned to be $0.1$), which could have a marked impact on sim2real performance if the target domain was significantly different. Investigating the generalization of various disturbance model choices to a wide variety of target domains would be interesting future work.
>
> We also included a comparison to DATT-L1 with DATT-RMA. Our adaptation network architecture consists of 3 1D convolutional layers with 64 channels each and a kernel size of 8 for each, followed by 3 fully connected layers of size 32 and ReLU activations. The input consists of the previous 50 state-action pairs. The adaptation network is trained by rolling out the same main DATT policy controller on line 2 of the above table, so direct comparison can be made.
>
> In simulation, DATT-L1 obtains a tracking error of $0.062 \pm 0.011$ while DATT-RMA obtains a tracking error of $0.055 \pm 0.009$ for tracking infeasible trajectories under wind, showing RMA does similar or better. However, from the table above, on the real drone, we see that the RMA adaptation network performs worse than L1 control, likely because the closed-loop nature of L1 guarantees fast disturbance estimation for any state-action pairs. As briefly discussed in the paper, we found that the adaptation network is highly susceptible to the domain shift in state-action pair input, resulting in a significant sim2real gap, though we believe that future work could potentially improve the performance of neural network based adaptation strategy. Ultimately, the DATT framework provides easy extension to any adaptation strategy as demonstrated.
>
> We plan to add these results along with discussion to our final submission.

---

> > ### Comment · Reviewer_fQDU · 2023-08-15
> > **on the time-varying disturbance**
> >
> > These new results look good to me. One thing I'd like to point out is that the new parameter \sigma does not introduce an extra parameter since the previous constant disturbance is needed for the same purpose.

---

### Decision · Program_Chairs · 2023-08-30

**Decision:**

Accept (Oral)

**Comment:**

## Summary of Paper
The paper combines ideas from control and RL to come up with quadrotor controllers that are robust to external disturbances. The controllers are trained in simulation and extensively evaluated on a real quadrotor.

## Summary of Reviews
The reviewers found the paper to be well written and interesting. They had a few questions about the details of the method and requested stronger baselines.

## Influence of Rebuttal
With the extensive new results and the clarifications all concerns of the reviewers have been addressed.

## ToDos for Final Version
Please incorporate all the material from the rebuttal in the final version and also see comment by Reviewer fQDU on \sigma.